# Evaluation of the Relationships between Intestinal Regional Lymph Nodes and Immune Responses in Viral Infections in Children

**DOI:** 10.3390/ijms23010318

**Published:** 2021-12-28

**Authors:** Yayoi Aoki, Tomoya Ikeda, Naoto Tani, Miho Watanabe, Takaki Ishikawa

**Affiliations:** 1Department of Legal Medicine, Osaka City University Medical School, Osaka 545-8585, Japan; ikeda.tomoya@med.osaka-cu.ac.jp (T.I.); tani.naoto@med.osaka-cu.ac.jp (N.T.); watanabe.miho2@med.osaka-cu.ac.jp (M.W.); takaki@med.osaka-cu.ac.jp (T.I.); 2Forensic Autopsy Section, Medico-Legal Consultation and Postmortem Investigation Support Center (MLCPI-SC), Osaka 545-8585, Japan; 3Laboratory of Clinical Regenerative Medicine, Department of Neurosurgery, Faculty of Medicine, University of Tsukuba, Health and Medical Science Innovation Laboratory 403, Tsukuba City 305-8575, Japan

**Keywords:** immune system, immunoglobulin A, interferon β, viral infection, regional lymph node, Peyer’s patch, child, cell culture

## Abstract

Viral infections increase the risk of developing allergies in childhood, and disruption of mucosal homeostasis is presumed to be involved. However, no study has reported a role for viral infections in such disruption. In this study, we clarified the mechanism of immunoglobulin A (IgA) overproduction in viral infections. Autopsies were performed on 33 pediatric cases, IgA and interferon (IFN)β levels were measured, and histopathological and immunohistochemical examinations were conducted. Furthermore, we cultured human cells and measured IFNβ and IgA levels to examine the effect of viral infections on IgA production. Blood IgA levels in viral infections were higher than in bacterial infections. Moreover, IFNβ levels in most viral cases were below the detection limit. Cell culture revealed increased IgA in gastrointestinal lymph nodes, especially in Peyer’s patches, due to enhanced IFNβ after viral stimulation. Conversely, respiratory regional lymph nodes showed enhanced IgA with no marked change in IFNβ. Overproduction of IgA, identified as an aberration of the immune system and resulting from excessive viral infection-induced IFNβ was observed in the intestinal regional lymph nodes, particularly in Peyer’s patches. Further, increased IgA without elevated IFNβ in the respiratory system suggested the possibility of a different mechanism from the gastrointestinal system.

## 1. Introduction

It is known that aberrations of the immune system, such as allergies, occur in childhood [1,2,3] and are presumed to involve the disruption of mucosal homeostasis. This is attributed to stimulation by foreign antigens of lymph nodes in the respiratory or gastrointestinal systems in childhood [4,5,6]. Particularly, viral infections and aberrations of the immune system, such as allergies, have recently been gaining attention. However, no definite correlation between viral infections and immune disorders has been demonstrated. Furthermore, it is unclear whether gastrointestinal or respiratory infections trigger immune disorders, including systemic inflammatory responses such as allergies [7,8], via the same mechanism.

Generally, interstitial cells in gastrointestinal regional lymph nodes express interferon (IFN)β, a type I interferon, upon stimulation by the bacterial flora. IFNβ alters normal dendritic cells into mucosal dendritic cells; these mucosal dendritic cells surface express A Proliferation Inducing Ligand and B cell Activating Factor, which are B cell activating molecules, causing B cells to differentiate into plasma cells so as to constantly produce immunoglobulin A (IgA). Recently, such excessive IgA production or IgA deficiency has drawn attention as a cause of aberrations of the immune system, including the development of type I allergies [9,10]. As with other immunoglobulins, IgA in the blood is derived from the bone marrow. However, the majority of IgA in the body is derived from mucosal regional lymph nodes and is usually distributed on mucosal surfaces as secretory IgA (sIgA). IgA on mucosal surfaces binds to antigens of various types, thereby functioning as a barrier against foreign antigens [11,12,13]. It is thought that the homeostatic environments surrounding regional lymph nodes in the gastrointestinal or respiratory system are disrupted by foreign antigen stimulation, including viral infections, affecting IgA secretion so that immunological aberrations, such as allergies, ultimately develop [14,15,16].

The present study aimed to investigate whether differences in location among various lymph nodes infected with viruses might affect IgA production. We focused on regional lymph nodes in the gastrointestinal and respiratory systems and performed blood biochemistry and histopathology on pediatric autopsy cases that had succumbed to viral or bacterial infections, with a view to the homeostatic IgA production mechanisms. Furthermore, we created viral infection models in cultured cells of various types of regional and non-regional lymph nodes and assessed IFNβ expression and subsequent IgA production in each lymph node.

## 2. Results

### 2.1. Comparisons between Clinical Reference Values and Values Obtained from Autopsies

We compared the clinical reference values of blood total IgA in childhood by age and those in the autopsy cases. The proportion of cases that showed higher total IgA levels than the clinical reference value was 54.5% (*n* = 6/11) in the viral infection group, 23.0% (*n* = 3/13) in the bacterial infection group, and 22.2% (*n* = 2/9) in the non-infection group.

#### 2.1.1. Blood Total IgA and sIgA Levels by Infection Category

Blood total IgA levels in the autopsy cases were compared among the viral infection, bacterial infection, and non-infection groups. The results showed that the viral infection group exhibited higher levels (25–150 mg/dL (median = 70 mg/dL)) than the bacterial infection group (11–130 mg/dL ((median = 28 mg/dL), *p* < 0.05) and non-infection group (16–78 mg/dL (median = 50 mg/dL), *p* < 0.05) (Figure 1a and Table 1).

As with total IgA levels, the viral infection group exhibited higher blood sIgA levels (14–3428 ng/mL (median = 141 ng/mL)) than the bacterial infection group (3–1240 ng/mL (median = 78 ng/mL), *p* = 0.401) and non-infection group (26–1051 ng/mL (median = 62 ng/mL), *p* = 0.342); however, no statistically significant differences were noted (Figure 1b and Table 1).

In terms of theoretical bone marrow-derived serum-type IgA level (calculated by subtracting sIgA from total IgA), the viral infection group exhibited higher levels (25.0–149.9 mg/dL (median = 70 mg/dL)) than the bacterial infection group (11.0–130.0 mg/dL (median = 28.0 mg/dL), *p* < 0.05) and non-infection group (16.0–78.0 mg/dL (median = 50.0 mg/dL), *p* < 0.05) (Figure 1c and Table 1).

#### 2.1.2. Blood IFNβ Levels by Infection Category

Blood IFNβ levels in autopsy cases exhibited different trends than total IgA and sIgA levels. Blood IFNβ levels were 1.2 pg/mL (i.e., the detection limit) or lower in 81.8% of the viral infection group (*n* = 9/11), 46% of the bacterial infection group (*n* = 6/13), and 77% of the non-infection group (*n* = 7/9) (Table 2).

No statistically significant difference in blood IFNβ levels was noted among the viral infection group (<1.2 to 4.24 pg/mL (median = 1.2 pg/mL)), bacterial infection group (<1.2 to 7.30 pg/mL (median = 1.2 pg/mL)), and non-infection group (<1.2 to 2.20 pg/mL (median = 1.2 pg/mL)).

### 2.2. Histopathology and Immunohistochemistry Findings

We investigated the histopathological changes in pulmonary hilar lymph nodes and small intestinal Peyer’s patches in the non-infection, bacterial infection, and viral infection groups. When observing pulmonary hilar lymph nodes by cause of infection, we noted intensive congestion in the non-infection group, but observed only a few edematous changes (Figure 2a(i); Table 3). The alveolar interstitia around the lymph nodes showed congestion and edematous changes (Figure 2a(i)). IFNβ immunostaining of pulmonary hilar lymph nodes in the non-infection group was “slightly positive” or “moderately positive” (Figure 2a(ii); Table 3), whereas the bronchial mucosae showed “strongly positive” staining. IgA immunostaining of pulmonary hilar lymph nodes in the non-infection group showed a “slightly to moderately positive” finding (Figure 2a(iii); Table 3), except for the tracheal epithelial cells, which showed “strongly positive” staining.

HE staining of intestinal Peyer’s patches in the non-infection group was “mostly negative” with respect to both congestion and edema (Figure 2a(iv); Table 3). IFNβ immunostaining of intestinal Peyer’s patches in the non-infection group was “mostly slightly positive” (Figure 2a(v); Table 3) and that of intestinal epithelial cells was “mostly slightly positive” (Figure 2a(v); Table 3). IgA immunostaining of intestinal Peyer’s patches in the non-infection group was “mostly slightly positive” (Figure 2a(vi); Table 3) and that of intestinal epithelial cells showed a “moderate or strong positive” finding.

HE staining of pulmonary hilar lymph nodes in the bacterial infection group showed “congested” and “edematous” findings (Figure 2b(i); Table 3). The alveolar interstitia around the lymph nodes presented with “congested,” “edematous,” and “hemorrhage into alveolae” findings. IFNβ immunostaining of pulmonary hilar lymph nodes in the bacterial infection group was “slightly positive” (Figure 2b(ii); Table 3). Virtually no IFNβ staining of the alveolar interstitia around the lymph nodes was observed. IgA immunostaining of pulmonary hilar lymph nodes in the bacterial infection group approached “moderate positivity” and the tracheal epithelial cells were “moderately positive” (Figure 2b(iii); Table 3). HE staining of intestinal Peyer’s patches in the bacterial infection group showed findings of “congestion” and “edema” as well as “mostly negative” or “slightly positive” findings (Figure 2b(iv); Table 3). IFNβ immunostaining of intestinal Peyer’s patches as well as epithelial cells in the bacterial infection group approached “slight positivity” (Figure 2b(v); Table 3). IgA immunostaining of intestinal Peyer’s patches in the bacterial infection group was “moderately positive” (Figure 2b(vi); Table 3) and that of intestinal epithelial cells was “moderately positive”.

HE staining of pulmonary hilar lymph nodes in the viral infection group revealed a “high level of congestion” and “edema” (Figure 2c(i); Table 3), and the alveolar interstitium around the lymph nodes showed a “high level of congestion” and “edema”. IFNβ immunostaining of pulmonary hilar lymph nodes in the viral infection group was “slightly positive” (Figure 2c(ii); Table 3). No IFNβ immunostaining was observed for the alveolar interstitia around the lymph nodes, and the bronchial mucosal epithelial cells were only slightly stained. IgA immunostaining of pulmonary hilar lymph nodes in the viral infection group were “moderately positive” or “strongly positive” (Figure 2c(iii); Table 3). The bronchial mucosal epithelial cells presented with “strongly positive” findings. HE staining of intestinal Peyer’s patches in the viral infection group did not show either “congestion” or “edema” (Figure 2c(iv); Table 3). IFNβ immunostaining of intestinal Peyer’s patches in the viral infection group was “moderately positive” or “strongly positive” (Figure 2c(v); Table 3), and staining of intestinal epithelial cells approached “strong positivity”. IgA immunostaining of intestinal Peyer’s patches in the viral infection group was “moderately positive” or “strongly positive” (Figure 2c(vi); Table 3), and staining of intestinal epithelial cells approached “moderate positivity” or were “strongly positive”.

### 2.3. Cell Culture and Cell Stimulation

#### 2.3.1. Lymph Node Interstitial Cell Stimulation

Different poly(I:C) concentrations were added to each lymph node interstitial cell type. As a result, an apparent elevation of IFNβ expression (1.97–2.35 pg/mL, average = 2.13 pg/mL) was noted with 10 ng/mL of poly(I:C) in intestinal Peyer’s patch (gastrointestinal regional lymph node) interstitial cells. IFNβ expression was elevated in a poly(I:C) concentration-dependent manner; the addition of 10,000 ng/mL of poly(I:C) resulted in levels of 15.18–22.06 pg/mL (average = 19.36 pg/mL).

In mesenteric lymph node interstitial cells, the addition of 1000 ng/mL of poly(I:C) elevated IFNβ levels (3.22–3.42 pg/mL, average = 3.34 pg/mL). The IFNβ level upon addition of 10,000 ng/mL of poly(I:C) was 7.26–7.56 pg/mL (average = 7.38 pg/mL). Likewise, in the fossa axillaris lymph node (non-gastrointestinal regional lymph node)-origin interstitial cells used as a control, the addition of 1000 ng/mL of poly(I:C) resulted in elevated IFNβ expression (2.23–2/41 pg/mL, average = 2.47 pg/mL); furthermore, the IFNβ level upon addition of 10,000 ng/mL was 5.94–6.33 pg/mL (average = 6.14 pg/mL). The rate of IFNβ elevation was high in intestinal Peyer’s patch (gastrointestinal regional lymph node) interstitial cells and low in fossa axillaris lymph node-origin interstitial cells (Figure 3a; Table 4).

Pulmonary hilar node-origin interstitial cells treated with 100 ng/mL poly (I:C) exhibited IFNβ expression levels of 1.27–1.33 pg/mL (average = 1.29 pg/mL). This level was approximately twice the IFNβ level in the absence of poly(I:C), 0.62–0.68 pg/mL (average = 0.65 pg/mL). Furthermore, IFNβ expression was elevated in a poly(I:C) concentration-dependent manner; the addition of 10,000 ng/mL of poly(I:C) resulted in IFNβ levels of 6.27–6.41 pg/mL (average 6.34 pg/mL). Poly(I:C) treatment of the inguinal lymph node (non-respiratory regional lymph node)-origin interstitial cells, used as a control, exhibited similar results to the pulmonary hilar node (respiratory regional lymph node)-origin interstitial cells. The IFNβ level upon addition of 100 ng/mL of poly(I:C) was 0.95–0.98 pg/mL (average = 0.97 pg/mL), which was approximately twice the IFNβ level in the absence of poly(I:C) (0.48–0.50 pg/mL, average = 0.49 pg/mL). IFNβ expression was elevated in a poly(I:C) concentration-dependent manner; the IFNβ expression level upon addition of 10,000 ng/mL of poly(I:C) was 5.72–5.86 pg/mL (average = 5.79 pg/mL) (Figure 3b and Table 4). However, the rates of IFNβ elevation in the pulmonary hilar and inguinal lymph nodes were both minor, and there was no remarkable difference in IFNβ expression between the respiratory regional and non-regional lymph nodes.

#### 2.3.2. Lymph Node Lymphocyte Stimulation

IgA production levels were measured after the addition of different IFNβ concentrations to lymphocytes from various types of lymph nodes. The addition of 50 ng/mL of IFNβ resulted in apparent increases in IgA production (33.5–35.5 ng/mL, average = 34.4 ng/mL) in intestinal Peyer’s patch (gastrointestinal regional lymph node) lymphocytes. Moreover, IgA production was increased in an IFNβ concentration-dependent manner; the addition of 10,000 ng/mL resulted in 146.6–153.3 ng/mL (average = 150.5 ng/mL). Similarly, IgA production in gastrointestinal regional lymph node lymphocytes also increased depending on the concentration of IFNβ. The addition of 50 ng/mL of IFNβ increased IgA production (12.6–13.9 ng/mL, average = 13.3 ng/mL), and the addition of 10,000 ng/mL IFNβ resulted in peak IgA levels, 35.7–38.3 ng/mL (average 37.2 ng/mL) (Figure 4a and Table 5). Conversely, lymphocytes of fossa axillaris lymph nodes, which are non-regional lymph nodes of the gastrointestinal tract, showed an extremely small IFNβ-induced increases in IgA production. However, the rate of increase was much lower (virtually no increase was noted) compared with that of the gastrointestinal regional lymph nodes; the IgA level without addition of IFNβ was 0.2 ng/mL (average 0.2 ng/mL), while the addition of 10,000 ng/mL of IFNβ resulted in an IgA level of <8.1 to 8.4 ng/mL (average = 8.2 ng/mL) (Figure 4a and Table 5).

Lymphocytes of pulmonary hilar lymph nodes, which are respiratory regional lymph nodes, produced 10.9–12.1 ng/mL (average = 11.5 ng/mL) of IgA upon addition of 50 ng/mL of IFNβ. This level was approximately 60 times the IgA level in the absence of poly(I:C), 0.2–0.3 ng/mL (average = 0.2 ng/mL). IgA production increased depending on the concentration of IFNβ, reaching 33.7–35.4 ng/mL (average = 34.4 ng/mL) upon addition of 10,000 ng/mL of IFNβ. In comparison, IgA levels in the inguinal lymph node (non-respiratory regional lymph node) lymphocytes, used as a control, after the addition of IFNβ increased depending on the concentration of IFNβ, similar to the results of respiratory regional lymph node lymphocytes; the IgA level upon addition of 50 ng/mL of IFNβ was 3.1–3.3 ng/mL (average = 3.2 ng/mL), which represented approximately 33 times the IgA level in the absence of IFNβ (0.1–0.2 ng/mL, average = 0.1 ng/mL). The IgA level reached 7.8–8.1 ng/mL (average = 7.9 ng/mL) upon addition of 10,000 ng/mL of IFNβ. However, the rate of increase was lower than that for pulmonary hilar lymph node (respiratory regional lymph node) lymphocytes (Figure 4b and Table 5).

#### 2.3.3. Morphological Changes upon Addition of IFNβ to Each Cultured Cell Type

As a result of the addition of IFNβ to lymphocytes, IFNβ concentration-dependent blast formation was noted in intestinal Peyer’s patch lymphocytes and pulmonary hilar lymph node lymphocytes (Figure 5a,b and Figure 6).

## 3. Discussion

The present study aimed to investigate whether differences in location between various lymph nodes infected with viruses might affect IgA production. Focusing on gastrointestinal and respiratory regional lymph nodes, we explored the behavior of IFNβ and IgA associated with viral infections in various lymph nodes. The results suggested that in gastrointestinal lymph nodes, including intestinal Peyer’s patches, IgA production increased via excessive IFNβ expression resulting from viral infection, thereby disrupting the homeostatic environment of lymph nodes to play a role in immunological aberrations such as allergies.

In autopsy cases where death was attributed to respiratory infection, viral infections exhibited higher blood total IgA and sIgA levels than in cases with bacterial infections. Given the fact that the majority of blood IgA is derived from bone marrow [17], the results showed that viral infections exhibited significantly higher levels than bacterial infections with respect to bone marrow-derived IgA, that is, total IgA minus sIgA. Since the clinical reference range of blood IgA varies considerably according to age, we conducted the evaluation by comparing blood IgA level in each autopsy case with the clinical reference range corresponding to their age. The evaluation also showed that many viral infection cases exhibited higher levels than the corresponding clinical reference range. Based on the data of the present study at this stage, the cutoff values for total IgA, sIgA, and serum IgA can be statistically estimated to be 50 mg/dL, 447 ng/mL, and 50 mg/dL, respectively; although, it is necessary to increase the number of samples and then calculate the cutoff values with a more accurate statistical analysis. Similarly, in past animal experiments, there has been a tendency for IgA levels to increase in immunological experiments; those studies are listed in Table 6 [18,19,20,21,22,23]. This study excluded cases of chronic infections (e.g., chronic hepatitis), malignant tumors, and blood dyscrasia (e.g., leukemia), as these diseases show increased IgA levels [24]. Moreover, patients prescribed steroids were excluded because steroid use is associated with low IgA levels [25]. Cases of hepatitis were also excluded because they may be prescribed IFN as a therapeutic agent [26]. Blood IFNβ levels below the detection limit were found in approximately 80% of cases in each of the viral infection, bacterial infection, and non-infection groups. Moreover, immunohistological investigation showed that, regarding the non-infection and bacterial infection groups, the IFNβ-positive rate was low in both pulmonary hilar and intestinal lymph nodes with the finding of marked congestion or hemorrhage noted. Furthermore, both the pulmonary hilar and intestinal lymph nodes showed weak IgA staining.

Meanwhile, there was marked congestion and edema in the alveolar interstitia in the viral infection group. Strong IFNβ immunostaining of the intestinal lymph nodes was observed, whereas the pulmonary hilar lymph nodes showed weak staining. IgA immunostaining indicated that both the pulmonary hilar and intestinal lymph nodes were strongly positive. Furthermore, blood IFNβ levels in the viral infection group of autopsy cases did not increase compared with those in the non-infection group. This is probably because the increase in IFNβ levels in the lymph nodes was not reflected in the blood IFNβ levels, demonstrating the fact that IFNβ expression associated with IgA production occurs in lymph nodes [9]. From the results of IFNβ and IgA immunostaining of respiratory and intestinal lymph nodes and measurement of blood IgA levels of the autopsy cases, it was suggested that IgA might be secreted without the involvement of IFNβ in viral infections.

The cell culture experiments using intestinal and respiratory lymph node lymphocytes and lymph node interstitial cells revealed that in intestinal Peyer’s patches (which are gastrointestinal regional lymph nodes), poly(I:C) stimulation caused IFNβ expression in the interstitial cells and that IgA production from lymph node lymphocytes increased in proportion to the amount of expressed IFNβ.

Poly(I:C) stimulation of respiratory lymph node interstitial cells produced low IFNβ levels that were virtually identical to that observed for inguinal lymph node (non-respiratory regional lymph node) interstitial cells. However, IFNβ stimulation of respiratory regional lymph node lymphocytes led to high levels of IgA secretion compared with that observed for inguinal lymph node lymphocytes, and the levels rapidly increased at 10 ng/mL of IFNβ in a concentration-dependent manner. That is, marked changes in IFNβ levels were not observed in respiratory regional lymph nodes with respect to viral infection stimulation, only IgA production resulting from IFNβ addition was increased in the respiratory regional lymph nodes. However, the number of intestinal infections (including viral and bacterial enteritis) in actual pediatric autopsy cases is extremely low; thus, it was not possible to investigate intestinal infections in the present study. Therefore, it was not possible to elucidate whether increases in IgA secretion without the involvement of IFNβ, as noted in the pulmonary hilar lymph nodes in the present study, might be observed in all viral infections.

Blast formation was observed in the cell culture experiment. Mature lymphocytes do not divide and proliferate, but when they encounter specific antigens, they adopt a pre-mature, juvenile cell format and start proliferating. This is called blast formation. Mature lymphocytes are classified as small lymphocytes, and juvenile lymphocytes are classified as large lymphocytes. Thus, the change in size observed in this experiment can be considered blast formation, an increase in the number of juvenile lymphocytes.

The results of this study suggested that activation of the IFNβ signaling pathway, as mentioned above, might selectively occur in intestinal regional lymph nodes, rather than occurring simultaneously in all lymph nodes. The reasons why intestinal Peyer’s patches, which are intestinal lymph nodes, selectively respond to IFNβ and IgA activities remain to be fully elucidated. However, one hypothesis assumes that the immune system of Peyer’s patches, which have the structural peculiarity of not being covered by a membrane (unlike other lymph nodes) and are aggregations of lymph nodules having germinal centers where antibodies are actively produced [27], is so well developed that immunological homeostasis in the absence of allergies is disrupted by childhood viral infections, resulting in allergy onset. The cell culture experiments were consistent with the results of IFNβ and IgA immunostaining of respiratory and intestinal lymph nodes obtained from autopsy cases, as well as with the finding of increases in childhood blood IgA levels associated with viral infections.

The present study revealed that enhanced IgA production, identified as an aberration of the immune system, in intestinal regional lymph nodes resulted from overexpression of IFNβ following viral infection stimulation, and IgA might be produced in the respiratory system via a different mechanism. The limitations of this study include the use of autopsy samples and the evaluation of monocultures, which does not allow understanding of the cell-to-cell relationship.

## 4. Materials and Methods

### 4.1. Autopsy Samples

A total of 33 pediatric autopsy cases (16 males and 17 females) aged 0–12 years (median, 0 years) was investigated. The case details are provided in Table 7. The number of cases of respiratory infections was 24, of which 11 were viral infections and 13 were bacterial infections. The following non-infection cases were investigated as controls: 2 cases of drowning; 2 cases of hyperthermia (heat stroke); 2 cases of fire fatality; 1 case of asphyxia; 1 case of acute circulation failure; and 1 case of blunt force injury. The following cases were excluded from the present study: cases of chronic infections (e.g., chronic hepatitis), cases of malignant tumors, cases of blood dyscrasia (e.g., leukemia), and cases prescribed any steroid known to be associated with low IgA levels from the perspective of clinical medicine. We performed PCR tests, cultures for identifying bacteria and viruses, and biochemical, histopathological and immunohistochemical examinations to distinguish among these infection groups. The non-infectious group was also confirmed to be free of infections by various viral and bacterial tests and histopathological examinations. These results were used to identify the group with infectious diseases. When multiple bacteria were detected, the main cause of death was considered one or several of them. However, since it is difficult to identify only one specific bacteria as the cause of death from an autopsy case, bacteria for which the bacteriologic culture examination results were indistinguishable from the bacteria found on micro-morphological examination of the lungs in the autopsy cases have been indicated with red lines and red words. Blood in the right heart was collected and centrifuged to obtain serum before tests were performed.

### 4.2. Measurement of Blood IgA and IFNβ Levels in Autopsy Cases

#### 4.2.1. Measurement of Blood Total IgA and sIgA Levels

Total blood IgA level was measured as follows: anti-human serum IgA was bound to 0.4 mL of a serum sample; the amount of IgA contained in the resulting complex was then measured in terms of turbidity at 694 nm using a turbidimetric immunoassay; and IgA levels in the serum sample were quantified. The reagent N-Assay TIA IgA-SH Nittobo (Nitto Boseki Co., Ltd., Tokyo, Japan) and an automatic analyzer JCA-BM8000 series (BM8030, Japan Electronics Industry Ltd., Tokyo, Japan) were used [28,29,30]. The measurement limits were 3–1100 mg/dL. The clinical reference range in healthy individuals was 110–410 mg/dL.

The sIgA enzyme-linked immunosorbent assay (ELISA) kit (#KR8870, Immunodiagnostik AG, Bensheim, Germany) [31,32,33] was used to measure sIgA levels in blood. The following measurement principle, method, and instrument were used: principle, sandwich ELISA; method, colorimetry; instrument, Multiskan FC Photometer (Thermo Fisher Scientific, Waltham, MA, USA). Measurements were performed at a wavelength of 450 nm. The measurement limits were 22.2–600 ng/mL, and the measurement sensitivity was 13.4 ng/mL.

#### 4.2.2. Measurement of Blood IFNβ Levels

Blood IFNβ levels were measured using an IFNβ ELISA kit (#41415-1, PBL Assay Science Co., Piscataway, NJ, USA) [34,35,36]. The following measurement principle, method, and instrument were used: principle, sandwich ELISA; method, colorimetry; instrument, Multiskan FC Photometer (Thermo Fisher Scientific). Measurements were performed at a wavelength of 450 nm. The measurement limits were 1.2–150 pg/mL.

### 4.3. Histopathology and Immunohistochemistry

Paraffin sections (4 μm) of the pulmonary hilar lymph nodes and the small intestine (duodenum or jejunum) including intestinal Peyer’s patches of the 33 autopsy cases were prepared and subjected to HE staining to investigate histopathological changes in the viral infection, bacterial infection, and non-infection groups.

Furthermore, immunostaining was performed on the paraffin-embedded sections from the 33 autopsy cases through color development of 3,3′-diaminobenzidine using an anti-human IgA rabbit polyclonal antibody (0.08 mg/mL, ×1, GTX20937, Fujifilm, Tokyo, Japan) and anti-human IFNβ rabbit polyclonal antibody (0.5 mg/mL, ×100, #ab140211, Abcam Co., Cambridge, UK) [37].

### 4.4. Cell Culture and Cell Stimulation

#### 4.4.1. Cell Culture System

In the experiments using cultured cells, Peyer’s patch interstitial cells and intestinal tract membrane lymph node-origin interstitial cells were used as gastrointestinal regional lymph node interstitial cells, and fossa axillaris lymph node (non-gastrointestinal regional lymph node)-origin interstitial cells were used as controls.

Intestinal Peyer’s patch lymphocytes and mesenteric lymph node lymphocytes were used as gastrointestinal regional lymph node lymphocytes, and fossa axillaris lymph node (non-gastrointestinal lymph node) lymphocytes were used as controls.

Pulmonary hilar node-origin interstitial cells were used as respiratory regional lymph node interstitial cells. Inguinal lymph node (non-respiratory regional lymph node)-origin interstitial cells were used as controls. Pulmonary hilar lymph node lymphocytes were used as respiratory regional lymph node lymphocytes. Inguinal lymph nodes (non-respiratory regional lymph node) were used as controls.

Cells used in the aforementioned experiments were provided from the Laboratory of Clinical Regenerative Medicine, Faculty of Medicine, and University of Tsukuba. The cells provided were originally lymphocytes and interstitial cells separated from various lymph nodes by Lymphocyte Separation Medium 1077 (#C-44010, Takara Bio Inc., Shiga, Japan). Therefore, they were not primary lymphocytes, but a cell line that was established by passaging those lymphocytes. Human lymph node interstitial cells and lymphocytes were cultured in 60 mm dishes containing 6 mL of Roswell Park Memorial Institute 1640 supplemented with human albumin for 72 h at 37 °C with a CO_2_ concentration of 5%.

#### 4.4.2. Lymph Node Interstitial Cell Stimulation Experiment

Peyer’s patch interstitial cells and intestinal tract membrane lymph node-origin interstitial cells were used as interstitial cells of gastrointestinal regional lymph nodes, and fossa axillaris lymph node-origin interstitial cells were used as controls. Pulmonary hilar node-origin interstitial cells were used as interstitial cells of respiratory regional lymph nodes, and inguinal lymph node-origin interstitial cells were used as controls.

Poly(I:C), which is a synthetic analogue of double-stranded ribonucleic acid (RNA) mimicking viral RNA and designed to simulate viral infections [38], was added to each cell type at a concentration of 0, 10, 100, 1000, or 10,000 ng/mL during cell culture. IFNβ was quantified by ELISA after 24 h.

#### 4.4.3. Lymphocyte Stimulation Experiment

Intestinal Peyer’s patch lymphocytes and mesenteric lymph node lymphocytes were used as gastrointestinal regional lymph node lymphocytes, and fossa axillaris lymph node lymphocytes were used as controls. Further, pulmonary hilar lymph node lymphocytes were used as respiratory lymph node lymphocytes, and inguinal lymph node lymphocytes were used as controls. IFNβ was added to each lymphocyte type at a concentration of 0, 10, 50, 100, 1000, or 10,000 ng/mL during lymphocyte culture. IgA levels were measured by ELISA after 24 h.

#### 4.4.4. Measurement of IFNβ and IgA Levels in Culture Media

IFNβ and IgA in the different cultured cell types were quantified using an ELISA kit (#41415-1, PBL Assay Science Co.) [34,35,36] and IgA ELISA kit (#ab196263, Abcam Co.), respectively [39,40].

### 4.5. Statistical Analyses

The nonparametric Mann–Whitney U test was used for comparisons between two groups. The Games–Howell test was used for comparisons between multiple factor groups. Analyses were performed using Microsoft Excel and IBM SPSS statistic viewer 24 [41,42].

## 5. Conclusions

The enhancement of IgA production, identified as an aberration of the immune system, in the gastrointestinal system is associated with increases in IFNβ expression triggered by viral infections. Moreover, IgA production in the respiratory system might be induced by a different mechanism.

## Figures and Tables

**Figure 1 ijms-23-00318-f001:**
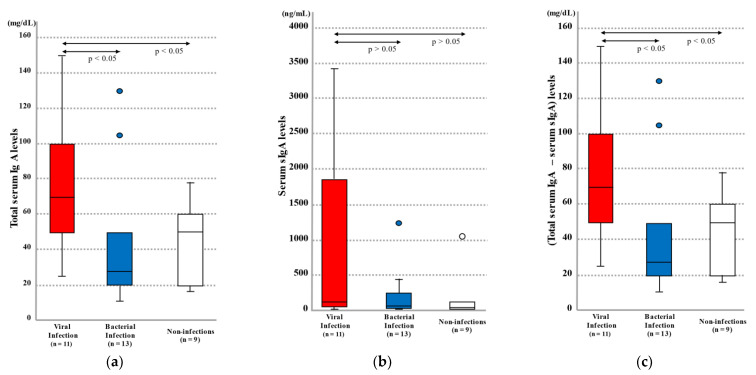
(**a**) Total serum IgA levels, (**b**) serum sIgA levels, and (**c**) total serum IgA levels–serum sIgA levels in autopsy cases by infection category. The lines in the graphs indicate median values and the lines above and below the boxes show 90% confidence intervals. The arrows indicate significant differences between two groups, at *p* < 0.05 according to the Mann–Whitney U test.

**Figure 2 ijms-23-00318-f002:**
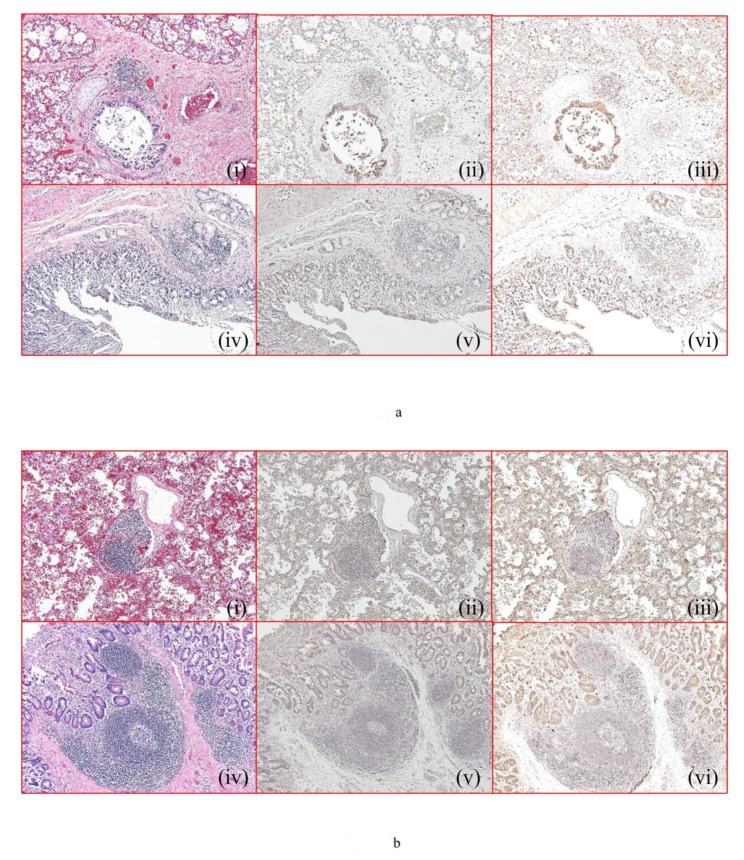
Histopathology of pulmonary hilar lymph nodes and intestinal Peyer’s patches in (**a**) non-infection, (**b**) bacterial infection, and (**c**) viral infection autopsy cases. Pulmonary hilar lymph node: (**i**) HE staining (×100), (**ii**) IFNβ immunostaining (×100), (**iii**) IgA immunostaining (×100). Intestinal Peyer’s patches: (**iv**) HE staining (×100), (**v**) IFNβ immunostaining (×100), (**vi**) IgA immunostaining (×100).

**Figure 3 ijms-23-00318-f003:**
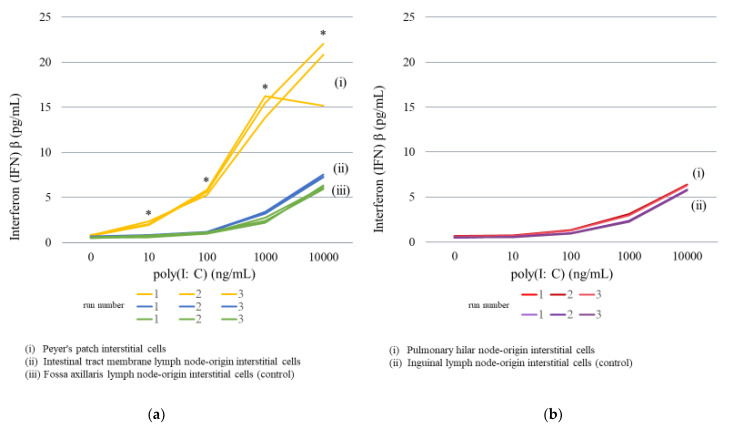
(**a**) Changes in IFNβ levels after addition of poly(I:C) to intestinal lymph node-related cultured cells: (**i**) Peyer’s patch interstitial cells, (**ii**) intestinal tract membrane lymph node-origin interstitial cells, (**iii**) fossa axillaris lymph node-origin interstitial cells (control). (**b**) Changes in IFNβ levels after addition of poly(I:C) to respiratory lymph node-related cultured cells by concentration: (**i**) Pulmonary hilar node-origin interstitial cells, (**ii**) inguinal lymph node-origin interstitial cells (control). * Indicates a significant difference between the groups, at *p* < 0.05 by the Games–Howell test.

**Figure 4 ijms-23-00318-f004:**
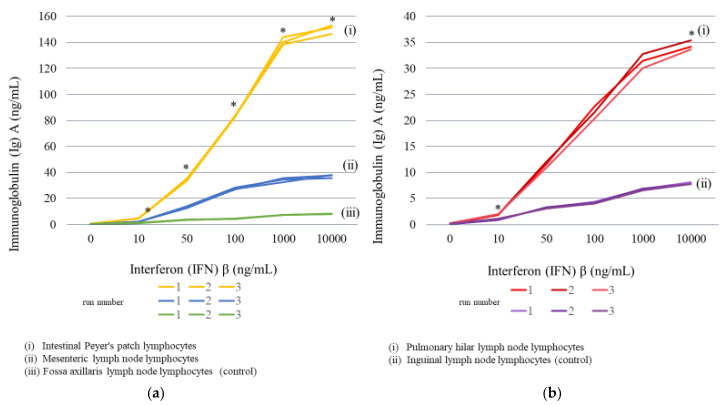
(**a**) Changes in IgA levels after addition of IFNβ to intestinal lymph node-related cultured cells: (**i**) Intestinal Peyer’s patch lymphocytes, (**ii**) mesenteric lymph node lymphocytes, (**iii**) fossa axillaris lymph node lymphocytes (control). (**b**) Changes in IgA levels after addition of IFNβ to respiratory lymph node-related cultured cells by concentration: (i) Pulmonary hilar lymph node lymphocytes, (**ii**) inguinal lymph node lymphocytes (control). * Indicates a significant difference between the groups, at *p* < 0.05 by the Games–Howell test.

**Figure 5 ijms-23-00318-f005:**
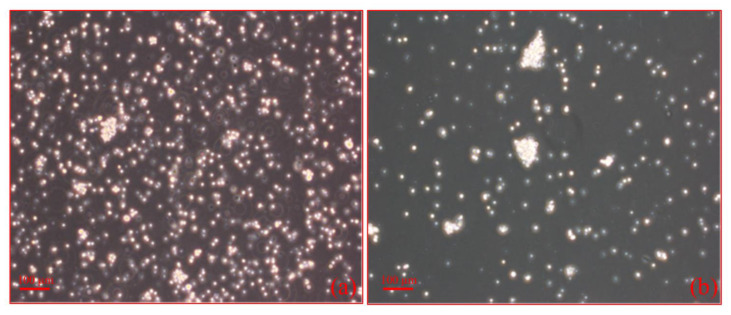
Cultured intestinal Peyer’s patch lymphocytes (**a**) before and (**b**) after the addition of 1000 ng/mL IFNβ.

**Figure 6 ijms-23-00318-f006:**
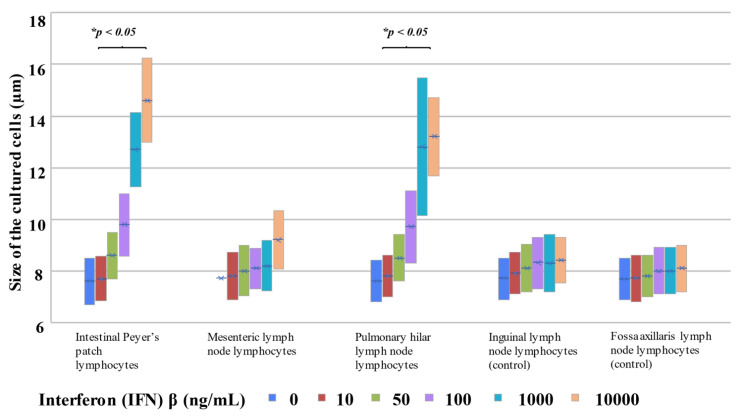
Cell sizes after the addition of IFNβ to lymphocytes in each type of regional lymph node. IFNβ concentration-dependent blast formation was observed in intestinal Peyer’s patch lymphocytes and pulmonary hilar lymph node lymphocytes.

**Table 1 ijms-23-00318-t001:** Total serum IgA levels, serum sIgA levels, and total serum IgA–serum sIgA levels by infection category.

Case No.	Age (Years)	Age (Months)	Sex	Cause of Death	Clinical ReferenceValue for TotalSerum IgA (mg/dL)	Total IgA (mg/dL) (Right Heart Blood)	sIgA (ng/mL)(Right Heart Blood)	Total IgA–sIgA (mg/dL) (Right Heart Blood)
V1	0	1	Male	Viral infection	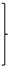	7–37	100	↑	2294.0	99.8
V2	0	1	Female	Viral infection	50	↑	3428.0	49.7
V3	0	2	Female	Viral infection	100	↑	1866.4	99.8
V4	0	3	Male	Viral infection	50	↑	52.9	50.0
V5	0	4	Male	Viral infection	25	→	50.0	25.0
V6	0	5	Male	Viral infection	16–50	150	↑	709.8	149.9
V7	0	10	Female	Viral infection	27–66	80	↑	53.7	80.0
V8	1	1	Male	Viral infection	36–79	50	→	141.3	50.0
V9	3	2	Female	Viral infection	27–246	64	→	419.3	64.0
V10	4	3	Female	Viral infection	29–256	70	→	17.1	70.0
V11	11	8	Female	Viral infection	42–295	94	→	14.0	94.0
B1	0	1	Male	Bacterial infection	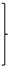	7–37	105	↑	116.3	105.0
B2	0	2	Male	Bacterial infection	20	→	118.6	20.0
B3	0	3	Male	Bacterial infection	40	↑	35.3	40.0
B4	0	3	Male	Bacterial infection	11	→	447.7	11.0
B5	0	4	Female	Bacterial infection	20	→	77.9	20.0
B6	0	4	Female	Bacterial infection	20	→	66.9	20.0
B7	0	5	Female	Bacterial infection	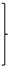	16–50	50	→	30.2	50.0
B8	0	7	Male	Bacterial infection	130	↑	96.7	130.0
B9	0	8	Female	Bacterial infection	25	→	48.2	25.0
B10	0	10	Female	Bacterial infection	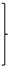	27–66	28	→	404.2	28.0
B11	0	10	Male	Bacterial infection	18	↓	1240.6	17.9
B12	1	3	Female	Bacterial infection	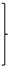	36–79	50	→	3.6	50.0
B13	1	5	Male	Bacterial infection	35	↓	54.5	35.0
O1	0	1	Male	Non-infection	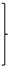	7–37	50	↑	1051.0	49.9
O2	0	2	Female	Non-infection	20	→	130.7	20.0
O3	0	3	Female	Non-infection	20	→	84.4	20.0
O4	0	4	Male	Non-infection	50	↑	158.5	50.0
O5	0	7	Female	Non-infection	16–50	50	→	45.1	50.0
O6	0	12	Male	Non-infection	27–66	50	→	41.4	50.0
O7	2	2	Female	Non-infection	27–246	70	→	27.4	70.0
O8	5	1	Male	Non-infection	29–256	78	→	26.3	78.0
O9	11	9	Female	Non-infection	42–295	16	↓	62.0	16.0

**Table 2 ijms-23-00318-t002:** Serum IFNβ levels by infection category.

Case No.	Age (Years)	Age (Months)	Sex	Cause of Death	IFNβ (pg/mL)
					(Right Heart Blood)
					(Cutoff Level < 1.2 pg/mL)
V1	0	1	Male	Viral infection	<1.2
V2	0	1	Female	Viral infection	<1.2
V3	0	2	Female	Viral infection	4.24
V4	0	3	Male	Viral infection	<1.2
V5	0	4	Male	Viral infection	<1.2
V6	0	5	Male	Viral infection	<1.2
V7	0	10	Female	Viral infection	<1.2
V8	1	1	Male	Viral infection	<1.2
V9	3	2	Female	Viral infection	3.90
V10	4	3	Female	Viral infection	<1.2
V11	11	8	Female	Viral infection	<1.2
B1	0	1	Male	Bacterial infection	<1.2
B2	0	2	Male	Bacterial infection	3.32
B3	0	3	Male	Bacterial infection	<1.2
B4	0	3	Male	Bacterial infection	<1.2
B5	0	4	Female	Bacterial infection	1.36
B6	0	4	Female	Bacterial infection	6.37
B7	0	5	Female	Bacterial infection	<1.2
B8	0	7	Male	Bacterial infection	<1.2
B9	0	8	Female	Bacterial infection	<1.2
B10	0	10	Female	Bacterial infection	3.21
B11	0	10	Male	Bacterial infection	7.30
B12	1	3	Female	Bacterial infection	4.40
B13	1	5	Male	Bacterial infection	2.17
O1	0	1	Male	Non-infection	<1.2
O2	0	2	Female	Non-infection	2.20
O3	0	3	Female	Non-infection	2.27
O4	0	4	Male	Non-infection	<1.2
O5	0	7	Female	Non-infection	<1.2
O6	0	12	Male	Non-infection	<1.2
O7	2	2	Female	Non-infection	<1.2
O8	5	1	Male	Non-infection	<1.2
O9	11	9	Female	Non-infection	<1.2

**Table 3 ijms-23-00318-t003:** Histopathology findings of pulmonary hilar lymph nodes and Peyer’s patch interstitial cells (−, negative; +, slight; ++, moderate; +++, strong) and results of lgA and IFNβ immunohistochemistry (1, slightly positive; 2, moderately positive; 3, strongly positive).

Case No.	Cause of Death	Lung	Gastrointestinal Tract	Lung		GASTROINTESTINAL TRACT	
		(Pulmonary Hilar	(Peyer’s Patch)	(Pulmonary Hilar		(Peyer’s Patch)	
		Lymph Node)		Lymph Node)			
		Congestion/Edema		Immunohistochemistry		Immunohistochemistry	
				IgA	IFNβ	IgA	IFNβ
V1	Viral infection	+++/++	+/−	1	1	2	3
V2	Viral infection	+++/++	+/−	3	1	3	1
V3	Viral infection	+++/+++	−/+	2	1	2	1
V4	Viral infection	+++/++	−/+	2	1	2	1
V5	Viral infection	+++/++	−/+	2	2	3	2
V6	Viral infection	+++/++	−/+	2	1	3	2
V7	Viral infection	++/++	+/−	2	1	3	2
V8	Viral infection	+++/++	−/−	2	1	3	2
V9	Viral infection	+++/++	+/−	2	1	3	2
V10	Viral infection	+++/++	+/−	3	1	3	2
V11	Viral infection	+++/++	+/−	2	2	2	2
B1	Bacterial infection	+/+	−/−	1	1	2	1
B2	Bacterial infection	+++/+++	+/++	2	2	1	1
B3	Bacterial infection	+/+	−/+	1	1	1	1
B4	Bacterial infection	+++/++	−/+	1	1	1	1
B5	Bacterial infection	+++/+++	−/−	2	1	2	1
B6	Bacterial infection	++/+++	+/+	1	2	2	1
B7	Bacterial infection	+/++	++/+	1	1	2	1
B8	Bacterial infection	++/+++	−/+	1	1	2	1
B9	Bacterial infection	++/+++	−/−	1	1	2	1
B10	Bacterial infection	+/++	−/+	1	1	1	2
B11	Bacterial infection	+++/++	−/+	1	1	2	1
B12	Bacterial infection	+/+	−/−	1	1	1	1
B13	Bacterial infection	+++/+++	−/−	1	1	2	1
O1	Non-infection	++/+	−/−	1	1	1	1
O2	Non-infection	+++/+	+/+	1	1	1	1
O3	Non-infection	+++/−	+/+	2	2	1	2
O4	Non-infection	++/+	−/−	1	2	1	1
O5	Non-infection	+/−	−/−	1	2	1	1
O6	Non-infection	+/−	−/+	1	1	1	1
O7	Non-infection	+++/+	−/−	1	1	1	1
O8	Non-infection	++/+	−/−	1	2	1	1
O9	Non-infection	+/−	−/−	1	1	1	1

**Table 4 ijms-23-00318-t004:** Amounts of IFNβ secreted from various lymph node interstitial cells after addition of different concentrations of poly(I:C).

Concentration of Poly(I:C)											
		0 (ng/mL)		10 (ng/mL)		100 (ng/mL)		1000 (ng/mL)		10,000 (ng/mL)	
Secretion of IFNβ (pg/mL)	Peyer’s patch		0.74		1.97		5.66		15.49		22.06
	interstitial cells		0.79		2.35		5.27		13.85		20.84
			0.85		2.06		5.85		16.23		15.18
		average	0.79	average	2.13	average	5.59	average	15.19	average	19.36
	Intestinal tract membrane		0.68		0.86		1.17		3.22		7.26
	lymph node-origin		0.72		0.75		1.06		3.42		7.56
	interstitial cells		0.54		0.83		1.21		3.39		7.33
		average	0.65	average	0.81	average	1.15	average	3.34	average	7.38
	Fossa axillaris lymph		0.56		0.62		1.02		2.78		6.14
	node-origin interstitial		0.51		0.73		0.98		2.23		6.33
	cells (control)		0.58		0.78		1.16		2.41		5.94
		average	0.55	average	0.71	average	1.05	average	2.47	average	6.14
	Pulmonary hilar		0.64		0.73		1.27		3.05		6.41
	node-origin		0.68		0.70		1.33		3.11		6.34
	interstitial cells		0.62		0.69		1.28		2.96		6.27
		average	0.65	average	0.71	average	1.29	average	3.04	average	6.34
	Inguinal lymph		0.48		0.55		0.97		2.33		5.86
	node-origin		0.48		0.57		0.98		2.35		5.78
	interstitial cells (control)		0.50		0.52		0.95		2.26		5.72
		average	0.49	average	0.55	average	0.97	average	2.31	average	5.79

**Table 5 ijms-23-00318-t005:** Amounts of IgA secreted from various lymph node interstitial cells after addition of different concentrations of IFNβ.

Concentration of IFNβ (ng/mL)													
		0 (ng/mL)		10 (ng/mL)		50 (ng/mL)		100 (ng/mL)		1000 (ng/mL)		10,000 (ng/mL)	
Secretion of IgA (ng/mL)	Intestinal		0.5		4.8		33.5		82.6		144.2		151.6
	Peyer’s patch		0.6		4.5		34.2		83.6		140.2		153.3
	lymphocytes		0.5		4.7		35.5		82.9		138.6		146.6
		average	0.5	average	4.7	average	34.4	average	83.0	average	141.0	average	150.5
	Mesenteric		0.3		2.2		13.4		26.8		35.8		37.7
	lymph node		0.3		2.1		12.6		27.2		32.6		38.3
	lymphocytes		0.4		2.2		13.9		28.4		34.8		35.7
		average	0.3	average	2.2	average	13.3	average	27.5	average	34.4	average	37.2
	Fossa axillaris lymph		0.2		1.1		3.4		4.6		7.3		8.4
	node lymphocytes		0.2		1.3		3.6		4.4		7.2		8.1
	(control)		0.2		1.2		3.8		4.2		7.4		8.2
		average	0.2	average	1.2	average	3.6	average	4.4	average	7.3	average	8.2
	Pulmonary hilar		0.2		1.8		11.6		22.7		31.5		34.2
	lymph node		0.2		1.9		12.1		21.6		32.8		35.4
	lymphocytes		0.3		2.0		10.9		20.4		30.1		33.7
		average	0.2	average	1.9	average	11.5	average	21.6	average	31.5	average	34.4
	Inguinal lymph node		0.1		0.9		3.2		4.4		6.8		8.1
	lymphocytes		0.1		0.9		3.3		4.2		6.9		7.8
	(control)		0.2		1.1		3.1		4.0		6.6		7.8
		average	0.1	average	1.0	average	3.2	average	4.2	average	6.8	average	7.9

**Table 6 ijms-23-00318-t006:** Review of the international literature on cases of animal experiments involving IgA production.

Case	Animal	Age (Week)	Sex	Preparation	Materials	Methods	Object	Results	Publication
1	mouse	8~12	female	intranasally infection of influenza virus	nasal wash	ELISA	IgA	↑	Sangster et al. 2003 [18]
2	mouse	6~8	female	subcutaneous immunizationof influenza virus	serum	ELISA	IgA	↑	Guy et al. 1998 [19]
								↓	
3	mouse	6~	male and female	oral inoculation and challenge of rotavirus	fecal pellet	ELISA	IgA	↑	Blutt et al. 2012 [20]
4	mouse	6	unidentified	canine distemper virus vaccination via intranasal route	intestinal mucus	ELISA	IgA	↑	Jiang et al. 2019 [21]
					genital tract wash			↑	
					nasal wash			↑	
					lungs			↑	
					fecal pellet			↑	
				canine distemper virus vaccination via oral route	intestinal mucus	ELISA	IgA	↑	
					genital tract wash			↑	
					nasal wash			↑	
					lungs			↑	
					fecal pellet			↑	
5	mouse	4~5	female	specific antigen administration	serum	Enzyme-linkedimmunoassay	IgA	↑	Frossard et al. 2004 [22]
					fecal pellet			→	
6	mouse	8~72	female	OVA administration and challenge	serum	ELISA	sIgA	↑	Santiago et al. 2008 [23]

↑ and ↓ indicate a significant increase and decrease, respectively, in IgA.

**Table 7 ijms-23-00318-t007:** Viral names and bacterial types detected in the (a) viral infection group and (b) bacterial infection group. (c) causes of death in the non-infection group.

(a) Viral Infection		(b) Bacterial Infection		(c) Other Causes of Death	
				(Non-Infection Cases)	
Case No.		Case No.		Case No.	
V1	Respiratory syncytial virus	B1	*Staphylococcus aureus*	O1	Heat stroke
	Influenza virus		*Acinetobacter *sp.	O2	Asphyxia
V2	Respiratory syncytial virus		*Citrobacter freundii*	O3	Acute circulation
V3	Respiratory syncytial virus	B2	*Mycoplasma pneumoniae*		failure
V4	Unidentified	B3	*Staphylococcus aureus*	O4	Drowning
V5	Respiratory syncytial virus		*Enterococcus *sp.	O5	Fire fatality
V6	Cytomegalovirus		*Mycoplasma pneumoniae*	O6	Heat stroke
V7	Adenoviridae		*Atopobium parvulum*	O7	Blunt force injury
V8	Cytomegalovirus		*Candida zeylanoides*		(contusion of the lung)
	Adenoviridae		*Enterococcus *sp.	O8	Drowning
V9	Respiratory syncytial virus		*Enterococcus durans*	O9	Fire fatality
	Adenoviridae		*Pseudomonas putida*		
V10	Adenoviridae		*Rhodotorula mucilaginosa*		
V11	Respiratory syncytial virus		*Trichosporon *sp.		
		B4	*Acinetobacter *sp.		
			*Escherichia coli*		
			*Enterococcus faecalis*		
			*Micrococcus *sp.		
		B5	*Mycoplasma pneumoniae*		
		B6	* Mycoplasma pneumoniae *		
			Methicillin-resistant		
			*Staphylococcus aureus (MRSA)*		
			*Escherichia coli*		
			*Klebsiella pneumoniae*		
		B7	* Streptococcus sp. *		
			* Klebsiella pneumoniae *		
			*Enterobacter cloacae*		
			*Bacillus cereus*		
			*Morganella morganii*		
			*Peptostreptococcus *sp.		
			*Porphyromonas asaccharolytica*		
			*Veillonella *sp.		
		B8	*Staphylococcus aureus*		
			Group B *Streptococcus*		
			*Pseudomonas aeruginosa*		
		B9	*Staphylococcus aureus*		
			*Streptococcus *sp.		
			*Acinetobacter *sp.		
			*Escherichia coli*		
			* Klebsiella pneumoniae *		
			*Citrobacter freundii*		
			*Candida *sp.		
			*Enterococcus faecalis*		
		B10	*Staphylococcus aureus*		
			*Streptococcus *sp.		
			* Escherichia coli *		
		B11	Methicillin-resistant		
			* Staphylococcus aureus (MRSA) *		
			* Bacillus cereus *		
		B12	* Klebsiella pneumoniae *		
		B13	*Streptococcus *sp.		
			*Escherichia coli*		
			*Mycoplasma pneumoniae*		
			* Staphylococcus aureus *		

#1-1. The bacteria underlined are the suspected pathogens. #1-2. In the non-infectious groups, bacterial, and viral identification tests and biochemical/histopathological examinations confirmed that there were no findings of infection.

## Data Availability

All the data are illustrated in the figures and tables.

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
