# Peer review of "Evaluation of the Relationships between Intestinal Regional Lymph Nodes and Immune Responses in Viral Infections in Children"

_ijms, 2021, doi:10.3390/ijms23010318_

Round 1

Reviewer 1 Report

The article " Evaluation of the relationships between intestinal regional lymph nodes and immune responses in viral infections in children." (ijms-1502429) by Aoki Y, et al. demonstrated that the different mechanism of increasing IgA between gastrointestinal and respiratory lymph nodes using sample from children received autopsy. This report mentioned that IgA level was increased in the children with viral compared with those with bacterial infection while INF-beta level was similar between these two groups. I considered that these data were very interesting, but there were several issues to be addressed as below.

  1. The patient characteristics were not clear. The author divided into viral infection, bacterial infection, and non-infection group. How did you identify with these infection groups? Was the cause of death several kinds of bacterial infection in a child, for example, all three bacterial infections, such as staphylococcus aureus, acinetobacter sp, and citrobacter freundii, was the cause of death for the case “B1”? Where was the infection site? Were there any evidences that the children of non-infection group was not infected exactly? I considered that the Peyer’s patch cell was not always infected by virus in the child diagnosed as viral infection because the incidence of gastrointestinal infection was very low in pediatric autopsy cases as you mentioned in discussion. Thus, the authors confirmed whether virus and /or bacteria infected in gastrointestinal and / or respiratory tract cells using PCR etc if possible.  

  1. What did “size of the cultured cells” mean in figure 6? Did the authors mean large cultured cells were identified as the lymphocyte with a lot of IgA in cytoplasm? The author should add your consideration in discussion.

  1. In methods, the author described “Cells used in the aforementioned experiments were provided from the Laboratory of Clinical Regenerative Medicine, Faculty of Medicine, and University of Tsukuba.” What were these cells? Were primary lymphocytes from gastrointestinal site and respiratory tract in human? The author should add the detail about cells used for experiments “cell culture and cell stimulation”.  

Author Response

Response to Reviewer 1 Comments

December 16, 2021

International Journal of Molecular Science

Dear Reviewer 1:

Thank you for reviewing our manuscript entitled “Evaluation of the relationships between intestinal regional lymph nodes and immune responses in viral infections in children”.

I would like to respond your comment.

Point 1: The patient characteristics were not clear. The author divided into viral infection, bacterial infection, and non-infection group. How did you identify with these infection groups?

Response 1: We performed PCR tests, bacterial and viral cultures, and biochemical, histopathological, and immunohistochemical examinations to distinguish among these infection groups. The non-infectious group was also confirmed to be free of infections by various viral and bacterial tests and histopathological examinations. These results were used to identify the group with infectious diseases.

Point 2: Was the cause of death several kinds of bacterial infection in a child, for example, all three bacterial infections, such as staphylococcus aureus, acinetobacter sp, and citrobacter freundii, was the cause of death for the case “B1”?

Response 2: #1. When multiple bacteria were detected, the main cause of death was one or several of them. However, it is difficult to identify specific bacteria as the cause of death from an autopsy case. Thus, for the bacteria for which the bacteriologic culture examination results agreed with the bacteria found on micro-morphological examination of the lungs in the autopsy cases, red font and an underline were used.

Point 3: Where was the infection site?

Response 3: All of the infectious disease cases in this study were respiratory tract infections, and this was confirmed by histopathological examination of most of the organs. Moreover, this was also confirmed by examination with lung-wiped fluid and bacterial and viral identification cultures of the lungs.

Point 4: Were there any evidences that the children of non-infection group was not infected exactly?

Response 4: In the non-infectious group, bacterial and viral identification tests and biochemical/histopathological examinations confirmed that there were no findings of infection.

Point 5: I considered that the Peyer’s patch cell was not always infected by virus in the child diagnosed as viral infection because the incidence of gastrointestinal infection was very low in pediatric autopsy cases as you mentioned in discussion. Thus, the authors confirmed whether virus and /or bacteria infected in gastrointestinal and / or respiratory tract cells using PCR etc if possible.

Response 5: We performed all of the pathological, biochemical, and microbiological examinations and diagnosed the respiratory infection cases that were included in this study. In addition, we confirmed histopathologically that we did not cause infections in the gastrointestinal tract. Therefore, samples were not collected for PCR.

Point 6: What did “size of the cultured cells” mean in figure 6? Did the authors mean large cultured cells were identified as the lymphocyte with a lot of IgA in cytoplasm? The author should add your consideration in discussion.

Response 6: #2. As described in the manuscript, we considered it blast formation. Mature lymphocytes do not divide and proliferate, but when they encounter specific antigens, they adopt a pre-mature, juvenile cell format and start proliferating. This is called blast formation. Mature lymphocytes are classified as small lymphocytes, and juvenile lymphocytes are classified as large lymphocytes, so that the change in size observed in this experiment can be considered blast formation, an increase in the number of juvenile lymphocytes. We added sentences about blast formation in the Discussion.

Point 7: In methods, the author described “Cells used in the aforementioned experiments were provided from the Laboratory of Clinical Regenerative Medicine, Faculty of Medicine, and University of Tsukuba.” What were these cells? Were primary lymphocytes from gastrointestinal site and respiratory tract in human? The author should add the detail about cells used for experiments “cell culture and cell stimulation”.

Response 7: #3. The cells provided were originally lymphocytes and interstitial cells separated from various lymph nodes by isolation medium. Therefore, they are not a primary lymphocyte line, but a cell line established by passaging those lymphocytes.

We added details about the cells provided.

#4. In addition, I changed the word "draining" to "regional" in the description of Figure 6.

Please feel free to contact me if you have any questions or require further information.

Reviewer 2 Report

In the present article, the authors study the mucosal disruption in viral infection in the autopsy samples. It was observed that IgA levels were increased in respiratory regional lymph nodes while insignificant change was observed in IFNβ levels. I have several reservations, my comments are appended as below:

  1. Do authors use any positive control while measuring IFNβ/ IgA levels?
  2. IHC- note the dilutions of antibodies used.
  3. Graphs- should be annotated with statistical inference.
  4. IHC- authors should attempt to quantify the expression.
  5. To have application in a clinical setting, which cutoff of IgA levels should be used? The authors should explain in the discussion section.
  6. Do authors sub-categorize the type of lymphocytes in profiling?
  7. The author should complement the findings with a competent animal model.

Author Response

Response to Reviewer 2 Comments

December 16, 2021

International Journal of Molecular Science

Dear Reviewer 2:

Thank you for reviewing our manuscript entitled “Evaluation of the relationships between intestinal regional lymph nodes and immune responses in viral infections in children”.

Point 1: Do authors use any positive control while measuring IFNβ/ IgA levels?

Response 1: Since samples were obtained by forensic autopsy, there was no serum that could be used as a positive control that certainly contained IgA and IFNβ. Therefore, we did not use IgA- and IFNβ-positive sera as positive controls, but we quantified them accurately using IgA and IFNβ standards in ELISA kits and their positive/negative controls.

Point 2: IHC- note the dilutions of antibodies used.

Response 2: #5. We have listed the concentrations before dilution and the dilution rate, though the diluted concentration had been listed.

Point 3: Graphs- should be annotated with statistical inference.

Response 3: #6. There was no mention of statistical inference in Figures 3 and 4. Statistical analysis was performed, and significant differences are marked with an asterisk (*).

Point 4: IHC- authors should attempt to quantify the expression.

Response 4: In legal medicine, immunohistochemical staining is mainly used to investigate the localization of proteins due to the marked individual differences in the morphology of organs and tissues obtained from autopsies, but it is difficult to quantify them for assessment. Thus, we quantified various serum IgA/IFNβ levels in this study.

Point 5: To have application in a clinical setting, which cutoff of IgA levels should be used? The authors should explain in the discussion section.

Response 5: #7. From the data of this study at this stage, we can estimate that the cutoff values for total IgA, sIgA, and serum IgA statistically are 50 mg/dL, 447 ng/mL, and 50 mg/dL, respectively, although we need to increase the number of samples and calculate the cutoff values using a more accurate statistical analysis.

Point 6: Do authors sub-categorize the type of lymphocytes in profiling?

Response 6: #8. We did not classify lymphocyte types in detail, but cells were separated using Lymphocyte Separation Medium 1077 (#C-44010, Takara Bio Inc., Shiga, Japan), and we considered that mononuclear cells were accurately collected containing lymphocytes.

We added information on the medium used for lymphocyte separation.

Point 7: The author should complement the findings with a competent animal model.

Response 7: #9. It is difficult to add data of animal experiments within 7 days.

Therefore, we added some references about the change in IgA production before and after the immunological experiment.

#4. In addition, I changed the word "draining" to "regional" in the description of Figure 6.

Please feel free to contact me if you have any questions or require further information.

Round 2

Reviewer 1 Report

I thanked the authors for polite response to my comments. I confirmed their response. However, their response was not reflected to this article. Therefore, the authors should add how to decide the patient characteristics, such as categorizing into the viral, bacteria, and non-infection groups, and detection of several concurrent bacterial infections in the method. 

Author Response

Response to Reviewer 1 Comments

December 20, 2021

International Journal of Molecular Science

Dear Reviewer 1

Thank you for re-reviewing our manuscript entitled “Evaluation of the relationships between intestinal regional lymph nodes and immune responses in viral infections in children”.

I would like to respond your comment.

Point 1: I thanked the authors for polite response to my comments. I confirmed their response. However, their response was not reflected to this article. Therefore, the authors should add how to decide the patient characteristics, such as categorizing into the viral, bacteria, and non-infection groups, and detection of several concurrent bacterial infections in the method.

Response 1: #1. Detailed information on the identification of infections and methods of grouping when multiple bacteria were detected has been added to the Materials and Methods section.

Please feel free to contact me if you have any questions or require further information.

Sincerely,

Yayoi Aoki

Reviewer 2 Report

I congratulate the authors for the modifications. I would suggest taking note of the following minor changes:

  1. Note the limitations of this study with clear heading.
  2. Specify in figure legend the type of t-test and meaning of *
  3. Table 6- is there any data on statistical inference available?

Author Response

Response to Reviewer 2 Comments

December 20, 2021

International Journal of Molecular Science

Dear Reviewer 2

Thank you for re-reviewing our manuscript entitled “Evaluation of the relationships between intestinal regional lymph nodes and immune responses in viral infections in children”.

I would like to respond your comment.

Point 1: Note the limitations of this study with clear heading.

Response 1: #2. The limitations of this study include the use of autopsy samples, resulting in the evaluation of monocultures, which does not allow understanding of the cell-to-cell relationship.

We have added the above sentence about this limitation in the Discussion section in the revised manuscript.

Point 2: Specify in figure legend the type of t-test and meaning of *

Response 2: #3. The types of statistical tests and what the “*” indicates have been added to the figure legends as recommended.

Point 3: Table 6- is there any data on statistical inference available?

Response 3: #4. Statistical inferences were added to the legend of Table 6.

Please feel free to contact me if you have any questions or require further information.

Sincerely,

Yayoi Aoki